# Adversarially Robust Decision Transformer

**Xiaohang Tang**[*]
University College London
xiaohang.tang.20@ucl.ac.uk

**Afonso Marques**[*]
University College London
afonso.marques.22@ucl.ac.uk

**Parameswaran Kamalaruban**[*]
Featurespace
kamal.parameswaran@featurespace.co.uk

**Ilija Bogunovic**
University College London
i.bogunovic@ucl.ac.uk

## Abstract

Decision Transformer (DT), as one of the representative Reinforcement Learning via Supervised Learning (RvS) methods, has achieved strong performance in offline learning tasks by leveraging the powerful Transformer architecture for sequential decision-making. However, in adversarial environments, these methods can be non-robust, since the return is dependent on the strategies of both the decision-maker and adversary. Training a probabilistic model conditioned on observed return to predict action can fail to generalize, as the trajectories that achieve a return in the dataset might have done so due to a suboptimal behavior adversary. To address this, we propose a worst-case-aware RvS algorithm, the Adversarially Robust Decision Transformer (ARDT), which learns and conditions the policy on in-sample minimax returns-to-go. ARDT aligns the target return with the worst-case return learned through minimax expectile regression, thereby enhancing robustness against powerful test-time adversaries. In experiments conducted on sequential games with full data coverage, ARDT can generate a maximin (Nash Equilibrium) strategy, the solution with the largest adversarial robustness. In large-scale sequential games and continuous adversarial RL environments with partial data coverage, ARDT demonstrates significantly superior robustness to powerful test-time adversaries and attains higher worst-case returns compared to contemporary DT methods.

## 1 Introduction

Reinforcement Learning via Supervised Learning (RvS) [8], has garnered attention within the domain of offline Reinforcement Learning (RL). Framing RL as a problem of outcome-conditioned sequence modeling, these methodologies have showcased strong performance in offline benchmark tasks [50, 53, 13, 37, 17]. One of the representative RvS methods is Decision Transformer (DT) [2], which simply trains a policy conditioned on target return via behavior cloning loss. Given the apparent efficacy and simplicity of RvS, our attention is drawn to exploring its performance in adversarial settings. In this paper, we aim to use RvS to achieve *adversarial robustness*, a critical capability for RL agents to manage environmental variations and adversarial disturbances [34, 42, 14, 46, 5, 36]. Such challenges are prevalent in real-world scenarios, e.g., changes in road conditions and decision-making among multiple autonomous vehicles.

In offline RL with adversarial actions, the naive training of action-prediction models conditioned on the history and observed returns such as DT, or expected returns such as ESPER and DoC [32, 54, 53], which we refer to as Expected Return-Conditioned DT (ERC-DT), may result in a non-robust policy.

---

[*]Equal Contribution.

38th Conference on Neural Information Processing Systems (NeurIPS 2024).

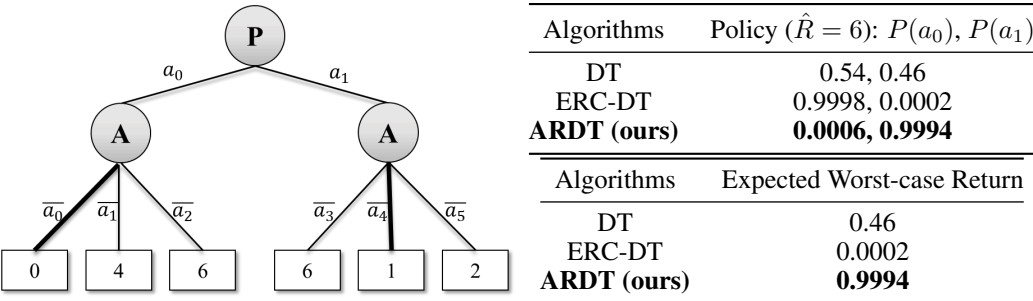

| Algorithms | Policy ($\hat{R} = 6$): $P(a_0)$, $P(a_1)$ |
|---|---|
| DT | 0.54, 0.46 |
| ERC-DT | 0.9998, 0.0002 |
| **ARDT (ours)** | **0.0006, 0.9994** |

| Algorithms | Expected Worst-case Return |
|---|---|
| DT | 0.46 |
| ERC-DT | 0.0002 |
| **ARDT (ours)** | **0.9994** |

Figure 1: **LHS** presents the game where decision-maker $P$ is confronted by adversary $A$. In the worst-case scenario, if $P$ chooses action $a_0$, $A$ will respond with $\bar{a}_0$, and if $P$ chooses $a_1$, $A$ will counter with $\bar{a}_4$. Consequently, the worst-case returns for actions $a_0$ and $a_1$ are 0 and 1, respectively. Therefore, the robust choice of action for the decision-maker is $a_1$. **RHS** displays tables of action probabilities and the worst-case returns for the Decision Transformer (DT), Expected Return-Conditioned DT (ERC-DT) methods and our algorithm, when conditioned on the largest return-to-go 6. After training using uniformly collected data that covered all possible trajectories, the results reveal that DT fails to select the robust action $a_1$, whereas our algorithm manages to do so.

This issue arises from the potential distributional shifts in the policy of the adversary during offline learning. Specifically, if the behavior policies used for data collection are suboptimal, a well-trained conditional policy overfitted to the training dataset may fail to achieve the desired target return during testing, if the adversary's policy at test time has changed or become more effective.

We illustrate this using a sequential game with data collected by a uniform random policy with full coverage shown in Figure 1. The standard RvS methods (DT and ERC-DT) when conditioned on high target return, show poor worst-case performance. DT's failure stems from training data containing the sequences $(a_0, \bar{a}_2)$ and $(a_1, \bar{a}_3)$, inaccurately indicating both actions $a_0$ and $a_1$ as optimal. ERC-DT missteps because the adversary's uniform random policy skews the expected return in favor of $a_0$ over $a_1$, wrongly favoring $a_0$. A robust strategy, however, exclusively selects $a_1$ against an optimal adversary to guarantee a high reward. Moreover, in more complex multi-decision scenarios, the adversarial strategy should not only minimize but also counterbalance with maximized returns by the decision-maker at the subsequent state, aligning with a *minimax* return approach for robust policy development.

**Main Contributions.** Consequently, there is a need for data curation to indicate the potential worst-case returns associated with each action in the Decision Transformer (DT), improving the robustness against adversarial actions. Building on this concept, we introduce the first robust training algorithm for Decision Transformer, the **Adversarially Robust Decision Transformer (ARDT)**. To the best of our knowledge, this paper represents the first exploration of the robustness of RvS methods from an *adversarial* perspective. In ARDT, we leverage Expectile Regression [29, 1] to transform the original returns-to-go to in-sample minimax returns-to-go, and train DT with the relabeled returns. In the experiments, we illustrate the robustness of ARDT in three settings (i) Games with full data coverage (ii) A long-horizon discrete game with partial data coverage (iii) Realistic continuous adversarial RL problems. We provide evidence showcasing the robustness of ARDT to more powerful adversary compared to the behavior policy and the distributional shifts of the adversary's policy in the test time. Furthermore, in the adversarial MuJoCo tasks, namely Noisy Action Robust MDP [42], ARDT exhibits better robustness to a population of adversarial perturbations compared to existing DT methods.

**Related work.** Reinforcement Learning via Supervised Learning is a promising direction for solving general offline RL problems by converting the RL problem into the outcome-conditioned prediction problem [17, 37, 10, 31, 8]. Decision Transformer (DT) is one of the RvS instance [2, 15], leveraging the strength of the sequence model Transformer [45]. The variants of DT have been extensively investigated to address the trajectory stitching [50, 53], stochasticity [31, 30, 32, 54], goal-conditioned problems [9], and generalization in different environments [26, 51, 21, 52]. For different problems, it is critical to select an appropriate target as the condition of the policy. $Q$-Learning DT [53] relabels the trajectories with the estimated $Q$-values learned to achieve trajectory stitching [18]. ESPER [32], aims at solving the environmental stochasticity via relabeling the trajectories with expected returns-to-go. Although our method also transform the returns-to-go to

estimated values, we instead associate worst-case returns-to-go with trajectories, aiming at improving its robustness against adversarial perturbations.

Adversarial RL can be framed as a sequential game; with online game-solving explored extensively in literature [19, 11, 40, 24, 20, 39, 38, 41]. In the offline setting, where there is no interaction with the environment, solving games is challenging due to distributional shifts in the adversary's policy, necessitating adequate data coverage. Tabular methods, such as Nash $Q$-Learning—a decades-old approach for decision-making in competitive games [12]—and other similar methods [4, 56] address these challenges through pessimistic out-of-sample estimation. While value-based methods are viable for addressing adversarial problems, our focus is on enhancing the robustness of RvS methods in these settings. Note that while some offline multi-agent RL methods, including DT-based approaches, are designed for cooperative games [26, 48, 44, 47], our research is oriented towards competitive games.

## 2 Problem Setup

We consider the adversarial reinforcement learning problem involving a protagonist and an adversary, as described by Pinto et al. [34]. This setting is rooted in the two-player Zero-Sum Markov Game framework [23, 33]. This game is formally defined by the tuple $(\mathcal{S}, \mathcal{A}, \bar{\mathcal{A}}, T, R, p_0)$. Here, $\mathcal{S}$ represents the state space, where $\mathcal{A}$ and $\bar{\mathcal{A}}$ denote the action spaces for the protagonist and adversary, respectively. The reward function $R : \mathcal{S} \times \mathcal{A} \times \bar{\mathcal{A}} \to \mathbb{R}$ and the transition kernel $T : \mathcal{S} \times \mathcal{A} \times \bar{\mathcal{A}} \to \Delta(\mathcal{S})$ depend on the state and the joint actions of both players. The initial state distribution is denoted by $p_0$. At each time-step $t \leq H$, both players observe the state $s_t \in \mathcal{S}$ and take actions $a_t \in \mathcal{A}$ and $\bar{a}_t \in \bar{\mathcal{A}}$. The adversary is considered adaptive if its action $\bar{a}_t$ depends on $s_t$ and $a_t$. The protagonist receives a reward $r_t = R(s_t, a_t, \bar{a}_t)$, while the adversary receives a negative reward $-r_t$.

Denote trajectory $\tau = (s_t, a_t, \bar{a}_t, r_t)_{t=0}^H$ and its sub-trajectory $\tau_{i:j} = (s_t, a_t, \bar{a}_t, r_t)_{t=i}^j$ for $0 \leq i \leq j \leq H$. The return-to-go refers to the sum of observed rewards from state $s_t$ onwards: $\widehat{R}(\tau_{t:H}) = \sum_{t'=t}^H r_{t'}$. The protagonist policy $\pi$ and the adversary policy $\bar{\pi}$ can be history-dependent, mapping history $(\tau_{0:t-1}, s_t)$ to the distribution over protagonist and adversary's actions, respectively.

In offline RL, direct interaction with the environment is unavailable. Instead, we rely on an offline dataset $\mathcal{D}$, which consists of trajectories generated by executing a pair of behavioral policies, $(\pi_{\mathcal{D}}, \bar{\pi}_{\mathcal{D}})$. Our objective in this adversarial offline RL setting is to utilize this dataset $\mathcal{D}$ to learn a protagonist policy that seeks to maximize the return, while being counteracted by an adversary's policy $\bar{\pi}$:

$$\max_\pi \min_{\bar{\pi}} \mathbb{E}_{\tau \sim \rho^{\pi, \bar{\pi}}} \left[ \sum_t r_t \right], \tag{1}$$

where $\rho^{\pi, \bar{\pi}}(\tau) = p_0(s_0) \cdot \prod_t \pi(a_t \mid \tau_{0:t-1}, s_t) \cdot \bar{\pi}(\bar{a}_t \mid \tau_{0:t-1}, s_t, a_t) \cdot T(s_{t+1} \mid s_t, a_t, \bar{a}_t)$. The maximin solution to this problem and its corresponding optimal adversary are denoted as $\pi^*$ and $\bar{\pi}^*$, respectively.

### 2.1 RvS in Adversarial Environments

We extend the RvS framework by incorporating adversarial settings within the Hindsight Information Matching (HIM) paradigm [9]. In adversarial HIM, the protagonist receives a goal (denoted by $z$) and acts within an adversarial environment. The protagonist's performance is evaluated based on how well it achieves the presented goal. Mathematically, the protagonist aims to minimize a distance metric $D(\cdot, \cdot)$ between the target goal $z$ and the information function $I(\cdot)$ applied to a trajectory $\tau$. This sub-trajectory follows a distribution $\rho$ dependent on both the protagonist's policy $\pi_z = \pi(a_t \mid \tau_{0:t-1}, s_t, z)$ and the test-time adversarial policy $\bar{\pi}_{\text{test}}$. Formally, the information distance is minimized as follows:

$$\min_\pi \mathbb{E}_{\tau \sim \rho^{\pi_z, \bar{\pi}_{\text{test}}}} \left[ D(I(\tau), z) \right], \tag{2}$$

where $\rho^{\pi_z, \bar{\pi}_{\text{test}}}$ represents the trajectory distribution obtained by executing rollouts in a Markov Game with the protagonist's policy $\pi_z$ and the test-time adversarial policy $\bar{\pi}_{\text{test}}$. Under HIM framework, Decision Transformer [2] uses adopt return-to-go value as the information function, while ESPER [32] employs the expected return-to-go. The well-trained RvS policy, including DT and ESPER, is derived

from behavior trajectories filtered based on the condition $I(\tau) = z$:

$$\pi_z = \pi(a_t \mid \tau_{0:t-1}, s_t, z) = \rho^{\pi_{\mathcal{D}}, \bar{\pi}_{\mathcal{D}}}(a_t \mid \tau_{0:t-1}, s_t, I(\tau) = z). \tag{3}$$

If the conditional policy ensures that the information distance in Eq. (2) is minimized to 0, the generated policy $\pi_z$ is robust against $\bar{\pi}_{\text{test}}$ by simply conditioning on a large target return, or even the maximin optimal protagonist $\pi^* = \lim_{z \to +\infty} \pi_z$ if the test-time adversary is optimal. Consequently, our objective shifts from learning a maximin policy in Eq. (1) to minimizing the distance in Eq. (2).

For a given target goal $z$, the trajectories where the information function $I(\tau) = z$ can be considered as optimal demonstrations of achieving that goal, i.e. information distance minimization, $D(I(\tau), z) = 0$. However, in adversarial environments, this information distance is hard to be minimized to 0. In adversarial offline RL, a trajectory in the dataset might achieve a goal due to the presence of a weak behavior adversary, rather than through the protagonist's effective actions. Filtering datasets for trajectories with high return-to-go could select the ones from encounters with suboptimal adversaries. Consequently, an agent trained through behavioral cloning on these trajectories is not guaranteed to achieve a high return when facing a different and powerful adversary at test time. We show this formally through a theorem:

**Theorem 1.** *Let $\pi_{\mathcal{D}}$ and $\bar{\pi}_{\mathcal{D}}$ be the data collecting policies used to gather data for training an RvS protagonist policy $\pi$. Assume $T(s_{t+1} \mid s_t, a_t, \bar{a}_t) = \rho^{\pi_{\mathcal{D}}, \bar{\pi}_{\mathcal{D}}}(s_{t+1} \mid \tau_{0:t-1}, s_t, a_t, \bar{a}_t, I(\tau) = z)$ [2], for any goal $z$ such that $\rho^{\pi_{\mathcal{D}}, \bar{\pi}_{\mathcal{D}}}(I(\tau) = z \mid s_0) > 0$, the information distance can be minimized: $\mathbb{E}_{\tau \sim \rho^{\pi_z, \bar{\pi}_{\text{test}}}}[D(I(\tau_t), z)] = 0$ if $\bar{\pi}_{\text{test}}(\bar{a}_t \mid \tau_{0:t-1}, s_t, a_t) = \rho^{\pi_{\mathcal{D}}, \bar{\pi}_{\mathcal{D}}}(\bar{a}_t \mid \tau_{0:t-1}, s_t, a_t, I(\tau) = z)$.*

Theorem 1 suggests that the protagonist policy $\pi_z$, defined in Eq. (3), can minimize the information distance when the test-time adversarial policy is $\bar{\pi}_{\text{test}} = \rho^{\pi_{\mathcal{D}}, \bar{\pi}_{\mathcal{D}}}(\bar{a}_t \mid \tau_{0:t-1}, s_t, a_t, I(\tau) = z)$. In other words, $\pi_z$, conditioned on a large target return, is guaranteed to be robust against $\rho^{\pi_{\mathcal{D}}, \bar{\pi}_{\mathcal{D}}}(\bar{a}_t \mid \tau_{0:t-1}, s_t, a_t, I(\tau) = z)$. Consequently, the policies of DT and ESPER are only guaranteed to perform well against behavior adversarial policies and will be vulnerable to powerful test-time adversaries since their information function is dependent on behavior adversary. This is a concern because, in practice, offline RL datasets are often collected in settings with weak or suboptimal adversaries.

In this context, to enhance robustness, we consider using an information function $I(\tau)$ to train the protagonist policy $\pi_z$ by simulating the worst-case scenarios we might encounter during test time. A natural choice for $I(\tau)$ is the **minimax return-to-go**, which we formally introduce in Section 3 and use as a central component of our method. With this $I(\tau)$ and sufficient data coverage, $\rho^{\pi_{\mathcal{D}}, \bar{\pi}_{\mathcal{D}}}(\bar{a}_t \mid \tau_{0:t-1}, s_t, a_t, I(\tau) = z)$ can be approximated to the optimal adversarial policy, implying that $\pi_z$ is robust against the optimal adversary. Furthermore, even without sufficient data coverage, $\pi_z$ remains more robust compared to previous DT methods, which tend to overfit to the behavior adversarial policy.

## 3 Adversarially Robust Decision Transformer

To tackle the problem of suboptimal adversarial behavior in the dataset, we propose **Adversarially Robust Decision Transformer (ARDT)**, a worst-case-aware RvS algorithm. In ARDT, the information function is defined as the expected minimax return-to-go:

$$I(\tau) = \min_{\bar{a}_{t:H}} \max_{a_{t+1:H}} \mathbb{E}_{\substack{s_{t'+1} \sim T(\cdot \mid s_{t'}, a_{t'}, \bar{a}_{t'}) \\ t' \in [t, H]}} \left[ \sum_{t'=t}^{H} R(s_{t'}, a_{t'}, \bar{a}_{t'}) \;\middle|\; \tau_{0:t-1}, s_t, a_t \right]. \tag{4}$$

Intuitively, this information function represents the expected return when an adversary aims to minimize the value, while the protagonist subsequently seeks to maximize it. To train a protagonist policy as described in Eq. (3), ARDT first relabels the trajectory with the minimax returns-to-go in Eq. (4). Secondly, the protagonist policy is then trained similarly to a standard Decision Transformer (DT). This two-step overview of ARDT is provided in Figure 2, where the left block includes the minimax return estimation with expectile regression for trajectory relabeling, and the right block represents the DT training conditioned on the relabeled returns-to-go.

---

[2]This condition can be readily met in environments formulated as a game with deterministic transitions, which are prevalent in many sequential games (e.g., Go and Chess [38]), as well as in frequently considered MuJoCo environments involving adversarial action noise [34, 3].

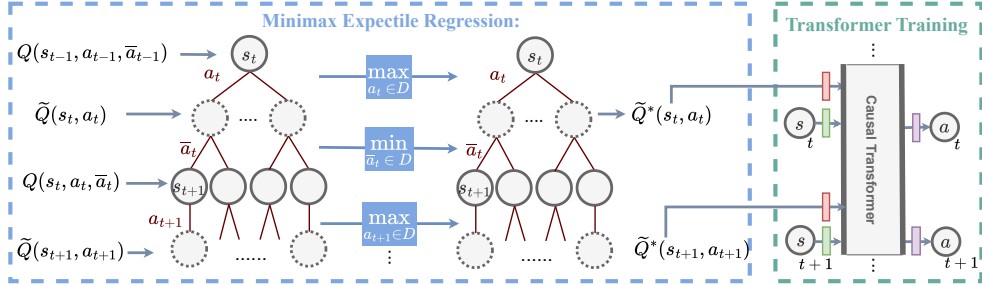

Figure 2: Training of Adversarially Robust Decision Transformer. We adopt Expectile Regression for estimator $\widetilde{Q}$ to approximate the in-sample minimax. In the subsequent protagonist DT training, we replace the original returns-to-go with the learned values $\widetilde{Q}^*$ to train policy.

We relabel the trajectories in our dataset with the worst-case returns-to-go by considering the optimal in-sample adversarial actions. In practice, we approximate the proposed information function using an (in-sample) *minimax returns-to-go* predictor, denoted by $\widetilde{Q}$. We simplify the notation of the appearance of actions in the dataset $a_t \in \mathcal{A} : \pi_{\mathcal{D}}(a_t|\tau_{0:t-1}, s_t) > 0$ and $\bar{a}_t \in \bar{\mathcal{A}} : \bar{\pi}_{\mathcal{D}}(\bar{a}_t|\tau_{0:t-1}, s_t, a_t) > 0$ to $a_t \in \mathcal{D}$ and $\bar{a}_t \in \mathcal{D}$, respectively. The optimal predictor $\widetilde{Q}^*$ is a history-dependent state-action value function satisfying the following condition that at any time $t = 1, \cdots, T$:

$$\widetilde{Q}^*(\tau_{0:t-1}, s_t, a_t) = \min_{\bar{a}_t \in \mathcal{D}} r_t + \mathbb{E}_{s_{t+1} \sim T(\cdot|s_t, a_t, \bar{a}_t)} \left[ \max_{a_{t+1} \in \mathcal{D}} \widetilde{Q}^*(\tau_{0:t}, s_{t+1}, a_{t+1}) \right], \qquad (5)$$

We adopt Expectile Regression (ER) [29, 1] to approximate $\widetilde{Q}$. This choice is particularly advantageous for RL applications because it helps avoid out-of-sample estimation or the need to query actions to approximate the minimum or maximum return [16]. Specifically, the loss function for ER is a weighted mean squared error (MSE):

$$L_{\text{ER}}^{\alpha}(u) = \mathbb{E}_u \left[ |\alpha - \mathbf{1}(u > 0)| \cdot u^2 \right]. \qquad (6)$$

Suppose a random variable $y$ follows a conditional distribution $y \sim \rho(\cdot|x)$, function $g_{\alpha}(x) := \arg\min_{g(x)} L_{\text{ER}}^{\alpha}(g(x) - h(x, y))$ can serve as an approximate minimum or maximum operator for the function $h(x, y)$ over all possible value of $y$, i.e.,

$$\lim_{\alpha \to 0} g_{\alpha}(x) = \min_{y:\rho(y|x)>0} h(x, y), \text{ or } \lim_{\alpha \to 1} g_{\alpha}(x) = \max_{y:\rho(y|x)>0} h(x, y). \qquad (7)$$

Our algorithm uses coupled losses for maximum and minimum operators to approximate the in-sample minimax returns-to-go. Formally, given an offline dataset $\mathcal{D}$, we alternately update the minimax and maximin returns-to-go estimators: $\widetilde{Q}_{\nu}(s, a)$ and $Q_{\omega}(s, a, \bar{a})$. In each iteration, we fix $\omega$ (or $\nu$), denoted as $\widehat{\omega}$ (or $\widehat{\nu}$), and then update $\nu$ (or $\omega$) based on the following loss functions:

$$\ell^{\alpha}(\nu) = \mathbb{E}_{\tau \sim \mathcal{D}} \left[ L_{\text{ER}}^{\alpha} \big( \widetilde{Q}_{\nu}(\tau_{0:t-1}, s_t, a_t) - Q_{\widehat{\omega}}(\tau_{0:t-1}, s_t, a_t, \bar{a}_t) \big) \right], \qquad (8)$$

$$\ell^{1-\alpha}(\omega) = \mathbb{E}_{\tau \sim \mathcal{D}} \left[ L_{\text{ER}}^{1-\alpha} \big( Q_{\omega}(\tau_{0:t-1}, s_t, a_t, \bar{a}_t) - \widetilde{Q}_{\widehat{\nu}}(\tau_{0:t}, s_{t+1}, a_{t+1}) - r_t \big) \right]. \qquad (9)$$

The minimizers ($\nu^*$ and $\omega^*$) of the two losses mentioned above (with $\alpha \to 0$), combined with Eq. (7), satisfy the following condition for all $s_t, a_t, \bar{a}_t \in \mathcal{D}$:

$$\widetilde{Q}_{\nu^*}(\tau_{0:t-1}, s_t, a_t) = \min_{\bar{a} \in \mathcal{D}} Q_{\omega^*}(\tau_{0:t-1}, s_t, a_t, \bar{a}), \qquad (10)$$

$$Q_{\omega^*}(\tau_{0:t-1}, s_t, a_t, \bar{a}_t) = \mathbb{E}_{s_{t+1} \sim \rho^{\pi_{\mathcal{D}}, \bar{\pi}_{\mathcal{D}}}(\tau_{0:t-1}, s_t, a_t, \bar{a}_t)} \left[ \max_{a' \in \mathcal{D}} \widetilde{Q}_{\nu^*}(\tau_{0:t}, s_{t+1}, a') + r_t \right]. \qquad (11)$$

By substituting Eq. (11) into Eq. (10), the minimax returns-to-go predictor $\widetilde{Q}_{\nu^*}$ satisfies Eq. (5).

The formal algorithm of ARDT is presented in Algorithm 1, where from line 3-6 is the minimax return estimation diagrammed as the left block in Figure 2, and line 7-10 is the Transformer training on

---

**Algorithm 1** Adversarially Robust Decision Transformer (ARDT)

---

1: **Inputs and hyperparameters:** offline dataset $\mathcal{D}$ containing collection of trajectories; $\alpha = 0.01$
2: **Initialization:** Initialize the parameters $\omega$ and $\nu$ for in-sample returns-to-go networks $Q_\omega$ and $\widetilde{Q}_\nu$ with the original returns-to-go, and parameter $\theta$ of the protagonist policy $\pi_\theta$.
    // expectile regression updates for in-sample returns-to-go networks
3: **for** iteration $k = 0, 1, \cdots$ **do**
4:     **for** trajectory $\tau \in \mathcal{D}$ **do**
        // update $\omega$ while keeping $\nu$ fixed
5:         $\omega \leftarrow \omega - \nabla_\omega \ell^{1-\alpha}(\omega)$, where $\ell^{1-\alpha}(\omega)$ is given in (9).
        // update $\nu$ while keeping $\omega$ fixed
6:         $\nu \leftarrow \nu - \nabla_\nu \ell^{\alpha}(\nu)$, where $\ell^{\alpha}(\nu)$ is given in (8).
    // relabeling dataset $\mathcal{D}$ using in-sample returns-to-go
7: Create $\mathcal{D}_{\text{worst}} = \left\{ \tau = (\cdots, \widetilde{R}_{t-1}, s_{t-1}, a_{t-1}, \widetilde{R}_t, s_t, a_t, \cdots) \mid \tau \in \mathcal{D}, \widetilde{R}_t = \widetilde{Q}_\nu(s_t, a_t) \right\}$
    // protagonist policy update
8: $L(\theta) = - \sum_{\tau \in \mathcal{D}_{\text{worst}}} \sum_t \log \pi_\theta(a_t \mid \tau_{0:t-1}, s_t, \widetilde{R}_t)$
9: **for** iteration $k = 0, 1, \cdots$ **do**
10:     $\theta \leftarrow \theta - \nabla_\theta L(\theta)$
11: **Output:** Policy $\pi_\theta$

---

the right of Figure 2. Initially, we warm up two returns-to-go networks with the original returns-to-go values from the dataset. This step ensures the value function converges to the expected values, facilitating faster maximin value estimation and guaranteeing accurate value function approximation at terminal states. Subsequently, we estimate the minimax returns-to-go by alternately updating the parameters of the two networks based on the expectile regression losses in Eq. (9) and Eq. (8). Once the minimax expectile regression converges, we replace the original returns-to-go values in the trajectories with the values predicted by $\widetilde{Q}_\nu$. Finally, we train the Decision Transformer (DT) conditioned on the states and relabeled returns-to-go to predict the protagonist's actions. Then the ARDT protagonist policy is ready to be deployed for evaluation in the adversarial environment directly.

## 4 Experiments

In this section, we conduct experiments to examine the robustness of our algorithm, Adversarially Robust Decision Transformer (ARDT), in three settings: (i) Short-horizon sequential games, where the offline dataset has full coverage and the test-time adversary is optimal (Section 4.1), (ii) A long-horizon sequential game, Connect Four, where the offline dataset has only partial coverage and the distributional-shifted test-time adversary (Section 4.2), and (iii) The standard continuous Mujoco tasks in the adversarial setting and a population of test-time adversaries (Section 4.3). We compare our algorithm with baselines including Decision Transformer and ESPER. The implementation details are in Appendix C. Notably, since the rewards and states tokens encompass sufficient information about the adversary's actions, all DT models are implemented not conditioned on the past adversarial tokens to reduce the computational cost.

### 4.1 Full Data Coverage Setting

The efficacy of our solution is first evaluated on three short-horizon sequential games with adaptive adversary: (1) a single-stage game, (2) an adversarial, single-stage bandit environment depicting a Gambling round [32], and (3) a multi-stage game. These are depicted in Figure 1, Figure 7, and Figure 9, respectively. The collected data consists of $10^5$ trajectories, encompassing all possible trajectories. The online policies for data collection employed by the protagonist and adversary were both uniformly random.

Figure 3 illustrates the efficacy of ARDT in comparison to vanilla DT and ESPER. Across all three environments, ARDT achieves the return of maximin (Nash Equilibrium) against the optimal adversary, when conditioned on a large target return. As illustrated in Figure 1, ARDT is aware of the worst-case return associated with each action it can take due to learning to condition on the relabeled

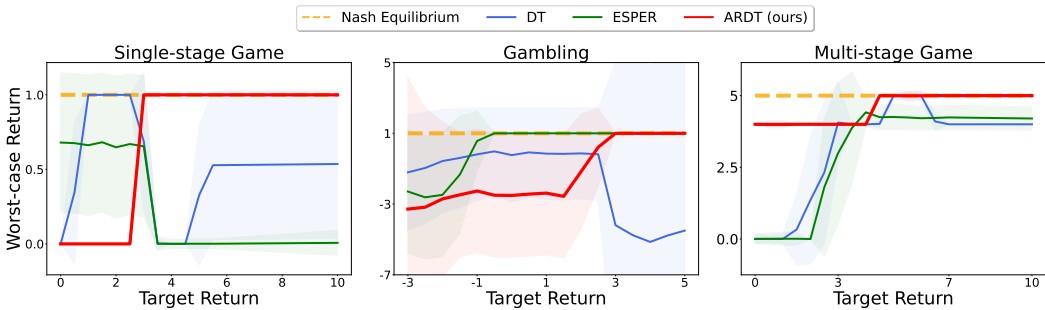

Figure 3: Worst-case return versus target return plot comparing the proposed ARDT algorithm against vanilla DT, on our Single-stage Game (left), Gambling (centre) and our Multi-stage Game (right), over 10 seeds.

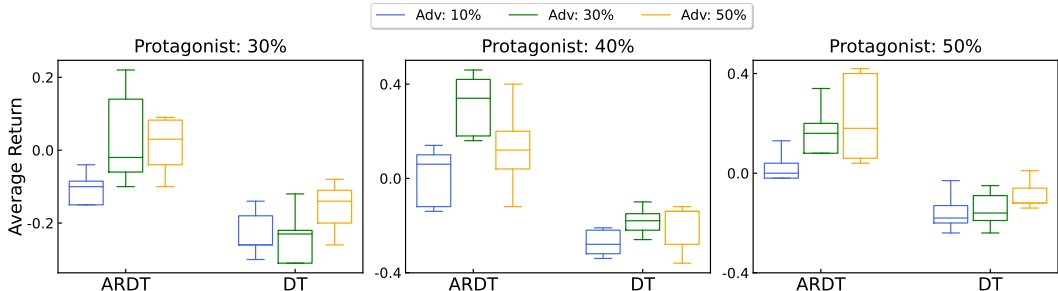

Figure 4: Average return of ARDT and vanilla DT on *Connect Four* when trained on suboptimal datasets collected with different levels of optimality for both the online protagonist's policy (30%, 40% and 50% optimal) and the adversary's policy (10%, 30%, 50% optimal), over 10 seeds. We test against a fixed adversary that acts optimally 50% of the time, and randomly otherwise.

minimax returns-to-go, and successfully takes robust action $a_1$. In single-stage game (Figure 3), ESPER is vulnerable to the adversarial perturbations, and DT fails to generate robust policy when conditioned on large target return. While, it is worth noting that DT has achieved worst-case return 1 when conditioning on target returns around 1. *This imposes the question: Can we learn the robust policy by simply conditioning on the largest worst-case return in testing with vanilla DT training?*

We show via the Gambling environment in Figure 3 that *vanilla DT can still fail even conditioning on the largest worst-case return.* In this environment, the worst-case returns of three protagonist actions are $-15$, $-6$ and $1$. At the target return $1$, DT is unable to consistently reach a mean return of $1$. Similar to DT in the single-stage game, it tends to assign equal likelihood to $a_1$ and $a_2$ to reach return $1$, leading to a drop of performance to less than $1$ under adversarial perturbations. Therefore, simply conditioning on the robust return-to-go in testing is insufficient to generate robust policy with vanilla DT. Moreover, as the target return approaches $5$, DT's performance drops more since DT tends to choose $a_0$ more often to hopefully achieve a return of $5$, but instead get negative return against the adversarial perturbations.

Conversely, ARDT's policy is robust since it is trained with worst-case returns-to-go. ESPER also performs well in Gambling due to that in this game the robust action is also the one with the largest expected return (Figure 9). ARDT also attains the highest return in a multi-stage game when conditioned on large target return, while other two algorithms fail. Consequently, in the context of full data coverage and an adaptive adversary, ARDT is capable of approximating the maximin (Nash Equilibrium) robust policy against the optimal adversarial policy, whereas DT is unable to do so.

## 4.2 Discrete Game with Partial Data Coverage

*Connect Four* is a discrete sequential two-player game with deterministic dynamics [32], which can also be viewed as a decision-making problem when facing an adaptive adversary. There is no

| Test-time adversary | 30% optimal | 50% optimal | 70% optimal | 100% optimal |
|---|---|---|---|---|
| DT | 0.18 (0.09) | -0.28 (0.05) | -0.57 (0.06) | -1 (0.0) |
| ESPER | 0.44 (0.09) | 0 (0.1) | -0.42 (0.05) | **-0.98 (0.01)** |
| **ARDT (ours)** | **0.55 (0.11)** | **0.11 (0.22)** | **0.02 (0.09)** | -1 (0.0) |

Table 1: Average returns of ARDT, vanilla DT and ESPER on *Connect Four* when trained on data mixed from both a near-random (30%, 50%) dataset and a near-optimal (90%, 90%) dataset. Evaluation is done against different optimality levels of adversary.

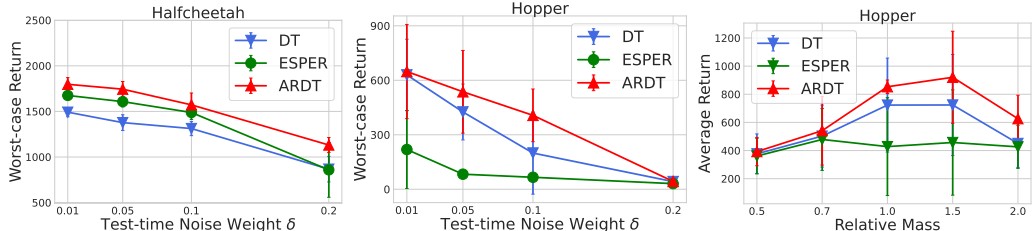

Figure 5: From left to right, (1) the worst-case return under adversarial perturbations with different weights in Halfcheetah, (2) in Hopper, and (3) average returns of algorithms in environments with different relative mass. The initial target returns in the environments Halfcheetah and Hopper are 2000 and 500, respectively.

intermediate reward; rather, there is a terminal reward of either $-1$, $0$, or $1$, meaning lose, tie or win for the first-moving player, respectively. Therefore we fix the target return in Connect Four as $1$. We fix the protagonist to be the first-move player. This game has a maximum horizon of 22 steps. Since we have a solver of this game, we manage to collect the data with $\epsilon$-greedy policy. We define the optimality of an $\epsilon$-greedy policy of protagonist or adversary as $(1 - \epsilon) \cdot 100\%$. Each dataset includes a total of $10^6$ number of steps for variable-length trajectories.

The results of training ARDT and DT on suboptimal datasets are presented in Figure 4. To demonstrate the stitching ability of the two methods, datasets collected by suboptimal protagonist and adversarial policies were chosen, namely at 30%, 40% and 50% optimality for the protagonist and 10%, 30%, 50% optimality for the adversary. The figures show that ARDT outperforms DT learning from suboptimal datasets, and more clearly so when tested against more powerful adversaries than the behavior adversary.

We hypothesise that one important factor accounting for the negative returns of DT in the setting in 4 may be the lack of long, winning trajectories in the dataset. To exclude this factor, in Table 1 we mix near-random and near-optimal datasets for training, and test our algorithms against different levels of adversary. Compared to ESPER and DT, ARDT still significantly outperforms DT, which suggests that even in the presence of trajectories with high returns in the data, training DT with original returns-to-go in this setting will limit its ability to extract a powerful and robust policy. In addition, the results show that ESPER outperforms DT, indicating relabeling trajectories with expected return can help the extracted policy be robust to distributionaly-shifted and more powerful test-time adversarial policies. ESPER conditioned on the expected return generates policy overfitted to the behavioral adversarial policy, which can fail under distributional shift. However, ARDT as a worst-case-aware algorithm, can achieve greater robustness at test time.

Therefore, in discrete environment with partial data coverage and adaptive adversary, ARDT is more robust to distributional shift and a more powerful test-time adversary.

### 4.3 Continuous Adversarial Environments

In this section, we test the algorithms on a specific instance of Adversarial Reinforcement Learning problems, namely Noisy Action Robust MDP (NR-MDP), which was introduced by Tessler et al. [42]. This includes three continuous control problems of OpenAI Gym MuJoCo, Hopper, Walker2D and

| | Worst-case Return | | |
| Datasets | DT | ESPER | ARDT |
| --- | --- | --- | --- |
| Hopper-lr | 112.1 (63.4) | 66.8 (5.5) | **477.9 (219.9)** |
| Hopper-mr | 330.2 (286.8) | 103.2 (11.7) | **482.2 (83.7)** |
| Hopper-hr | **298.62 (319.72)** | 63.0 (11.5) | **331.69 (187.18)** |
| Walker2D-lr | **429.6 (31.9)** | **380.6 (156.9)** | **405.6 (39.8)** |
| Walker2D-mr | 391.0 (20.6) | 426.9 (29.7) | **508.4 (61.8)** |
| Walker2D-hr | 413.22 (54.69) | **502.6 (31.3)** | **492.63 (109.28)** |
| Halfcheetah-lr | 1416.0 (48.5) | **1691.8 (63.0)** | 1509.2 (33.8) |
| Halfcheetah-mr | 1229.3 (171.1) | **1725.6 (111.8)** | **1594.6 (137.5)** |
| Halfcheetah-hr | 1493.03 (32.87) | **1699.8 (32.8)** | **1765.71 (150.63)** |

Table 2: Results of DT, ESPER, and our algorithm ARDT with and without conditioned on past adversarial tokens: the **Worst-case Return** against 8 different online adversarial policies on MuJoCo *Noisy Action Robust MDP* tasks across 5 seeds. We tested with the same set of target returns and select the best target for each method. To expand the data coverage, we inject random trajectories collected by 0.1-greedy online policies into the pre-collected online datasets with high robust returns. We use low (suffix -lr), medium (suffix -mr) and high randomness (suffix -hr) to indicate the proportion of random trajectories.

Halfcheetah [43] involving the adversarial action noise. In NR-MDP, the protagonist and adversary act jointly. Specifically, they exact a total force over one or more parts of the body of the agent in a Mujoco environment, corresponding to the weighted sum of their individual forces (adversary's weight $\delta$). Despite no longer being in the discrete setting, the return minimax expectile regression in ARDT is implemented with the raw continuous states and actions input, without any discretization.

To collect the data, we first trained Action Robust Reinforcement Learning (ARRL) online policies on NR-MDP tasks [42] to create the online protagonist policy and adversary policy for evaluation. Subsequently, the pre-collected data consist of the replayed trajectories removing the ones with low cumulative rewards. We collect the suboptimal trajectories with the $\epsilon$-greedy of the saved ARRL agents, i.e. online protagonist and adversary policies. We then mix them with the pre-collected datasets to create the final training datasets. The proportion of random trajectories to pre-collected trajectories are $0.01 : 1$, $0.1 : 1$ and $1 : 1$, respectively, for datasets categorised as being low, medium, and high randomness. At test time, we evaluate each of our trained ARDT protagonist policies against a population of 8 adversarial online policies, each across 5 seeds. We sweep over the same set of target returns for all algorithms, and pick the best results.

As shown in Table 4, ARDT has overall better worst-case performance than DT and ESPER. In Hopper, ARDT has significantly higher worst-case return than both ESPER and DT. In Halfcheetah, ARDT has significantly higher worst-case return than DT and competitive results with ESPER. This is despite the much lower data coverage than in the previous settings due to the continuous environment. Low data coverage is in theory a greater limitation for ARDT given that it attempts to learn a single best worst-case outcome, rather than an expectation over outcomes. Even so, our proposed solution outperforms the baselines in most settings across all three environments. In addition, ARDT also has better performance when varying the adversary's weight $\delta$ and relative mass (Figure 5), implying the robustness of ARDT to test-time environmental changes. Therefore, in the context of a continuous state and action space, and when facing a diverse population of adversaries or environmental changes, ARDT trained with minimax returns-to-go can still achieve superior robustness than our baselines.

## 4.4 Ablation Study

Finally, we offer abalation study on the expectile level $\alpha$ in discrete environments. Specifically, we varied $\alpha$ for relabeling in Single-stage Game, Multi-stage Game and Connect Four. Then we tested against the optimal adversary in the first two environments and an $\epsilon$-greedy adversary ($\epsilon = 0.5$) for the third environment.

According to the first two environments, we can confirm a pattern consistent with our theory that smaller alpha leads to more robustness. According to the Equation (6)-(9) in the paper, it should

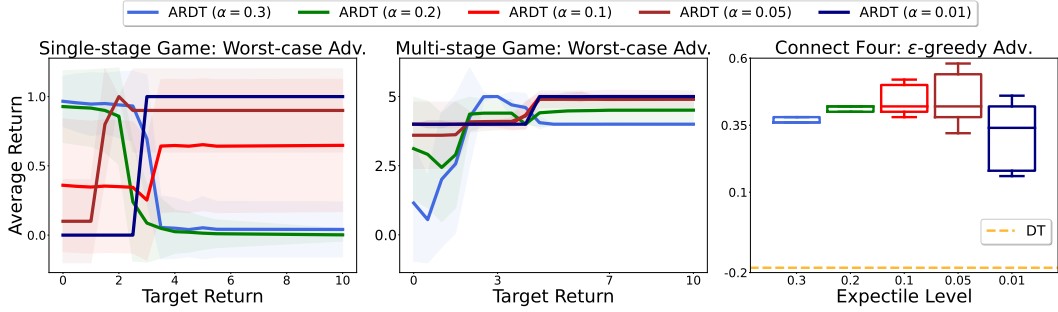

Figure 6: **Ablation study** on expectile level $\alpha$ over 10 random seeds. In connect four, we set the target return to be 1, and the adversary to be $\epsilon$-greedy ($\epsilon = 0.5$), i.e. taking uniform random policy with probability 0.5. On Connect Four, ARDT with $\alpha = 0.01$ acts too conservative.

be that when the $\alpha$ is closer to 0, our estimate of minimax return is more accurate, and thus our algorithm is more robust. When the alpha is increased to 0.5, our method is converted to Expected-Return-Conditioned Decision Transformer (e.g. ESPER) since the expectile loss in this case reduces to Mean Square Error. This can be confirmed by comparing the results of Single-stage game and Multi-stage game in Figure 6 and 3.

In the third environment, Connect Four, the performance initially increases as decreases, but eventually drops. This is due to the weak stochastic adversary. When the value is too small, the algorithm becomes overly conservative. However, ARDT still outperforms DT significantly. Additionally, it also implies that we can tune the parameter alpha to adjust the level of conservativeness

## 5 Conclusions and Limitations

This paper introduces a worst-case-aware training algorithm designed to improve the adversarial robustness of the Decision Transformer, namely Adversarially Robust Decision Transformer (ARDT). By relabeling trajectories with the estimated in-sample minimax returns-to-go through expectile regression, our algorithm is demonstrated to be robust against adversaries more powerful than the behavior ones, which existing DT methods cannot achieve. In experiments, our method consistently exhibits superior worst-case performance against adversarial attacks in both gaming environments and continuous control settings.

**Limitations.** In our experiments, our practical algorithm estimates the minimax returns-to-go without accounting for stochasticity, as the transitions in all tested environments, as well as many real-world game settings, are deterministic when conditioned on the actions of both players. Future work should explore both stochastic and adversarial environments to further evaluate the performance of our proposed solution.

**Broader Impact.** Adversarial RvS holds potential for improving the robustness and security of autonomous agents in dynamic and potentially hostile environments. By training agents to withstand adversarial actions and unpredictable conditions, our work can improve the reliability and safety of technologies such as autonomous vehicles, cybersecurity defense systems, and robotic operations.

## 6 Acknowledgments

Ilija Bogunovic was supported by the EPSRC New Investigator Award EP/X03917X/1; the Engineering and Physical Sciences Research Council EP/S021566/1; and Google Research Scholar award. Xiaohang Tang was supported by the Engineering and Physical Sciences Research Council [grant number EP/T517793/1, EP/W524335/1]. The authors would like to thank the Department of Statistical Science, in particular Chakkapas Visavakul, for co-ordinating the computer resources, and the Department of Electronic and Electrical Engineering, and Department of Computer Science at University College London for providing the computer clusters.

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

# Appendix

## A Proofs

**Theorem.** *1 Let $\pi_{\mathcal{D}}$ and $\bar{\pi}_{\mathcal{D}}$ be the data collecting policies used to gather data for training an RvS protagonist policy $\pi$. Assume $T(s_{t+1} \mid s_t, a_t, \bar{a}_t) = \rho^{\pi_{\mathcal{D}}, \bar{\pi}_{\mathcal{D}}}(s_{t+1} \mid \tau_{0:t-1}, s_t, a_t, \bar{a}_t, I(\tau) = z)$, for any goal $z$ such that $\rho^{\pi_{\mathcal{D}}, \bar{\pi}_{\mathcal{D}}}(I(\tau) = z \mid s_0) > 0$, the information distance can be minimized: $\mathbb{E}_{\tau \sim \rho^{\pi_z}, \bar{\pi}_{\text{test}}}[D(I(\tau), z)] = 0$ if $\bar{\pi}_{\text{test}}(\bar{a}_t \mid \tau_{0:t-1}, s_t, a_t) = \rho^{\pi_{\mathcal{D}}, \bar{\pi}_{\mathcal{D}}}(\bar{a}_t \mid \tau_{0:t-1}, s_t, a_t, I(\tau) = z)$.*

*Proof.* According to Eq. (3):

$$\pi(a_t \mid \tau_{0:t-1}, s_t, z) = \rho^{\pi_{\mathcal{D}}, \bar{\pi}_{\mathcal{D}}}(a_t \mid \tau_{0:t-1}, s_t, I(\tau) = z). \tag{12}$$

If the sufficient condition is satisfied: $\bar{\pi}_{\text{test}}(\bar{a}_t \mid \tau_{0:t-1}, s_t, a_t) = \rho^{\pi_{\mathcal{D}}, \bar{\pi}_{\mathcal{D}}}(\bar{a}_t \mid \tau_{0:t-1}, s_t, a_t, I(\tau) = z)$, combined with our assumption, we have

$$\bar{\pi}_{\text{test}}(\bar{a}_t \mid \tau_{0:t-1}, s_t, a_t) \cdot T(s_{t+1} \mid s_t, \bar{a}_t) = \rho^{\pi_{\mathcal{D}}, \bar{\pi}_{\mathcal{D}}}(s_{t+1} \mid \tau_{0:t-1}, s_t, a_t, I(\tau) = z). \tag{13}$$

Then, we have:

$$\mathbb{E}_{\tau \sim \rho^{\pi_z}, \bar{\pi}_{\text{test}}(\cdot \mid s_0)}[D(I(\tau), z)]$$
$$= \sum_{\tau} \rho^{\pi_z, \bar{\pi}_{\text{test}}}(\tau \mid s_0) \cdot D(I(\tau), z)$$
$$= \sum_{\tau} \prod_t \pi(a_t \mid \tau_{0:t-1}, s_t, z) \cdot \bar{\pi}_{\text{test}}(\bar{a}_t \mid \tau_{0:t-1}, s_t, a_t) \cdot T(s_{t+1} \mid s_t, a_t, \bar{a}_t) \cdot D(I(\tau), z)$$
$$= \sum_{\tau} \prod_t \rho^{\pi_{\mathcal{D}}, \bar{\pi}_{\mathcal{D}}}(a_t \mid \tau_{0:t-1}, s_t, I(\tau) = z) \cdot \rho^{\pi_{\mathcal{D}}, \bar{\pi}_{\mathcal{D}}}(s_{t+1} \mid \tau_{0:t-1}, s_t, a_t, I(\tau) = z) \cdot D(I(\tau), z)$$
$$= \sum_{\tau} \rho^{\pi_{\mathcal{D}}, \bar{\pi}_{\mathcal{D}}}(\tau \mid s_0, I(\tau) = z) \cdot D(I(\tau), z)$$
$$= \mathbb{E}_{\tau \sim \rho^{\pi_{\mathcal{D}}, \bar{\pi}_{\mathcal{D}}}(\cdot \mid s_0, I(\tau) = z)}[D(I(\tau), z)]$$
$$= 0.$$

The second equation satisfies by simply decomposing $\rho$ as in section 2. The third equation satisfies due to Eq. (13) and Eq. (3). The fourth equation satisfies by simply merging all $\rho$ terms. The final equation satisfies since the expectation in the last but one equation is over the probability distribution conditional on $I(\tau) = z$, leading to that for any sampled trajectory $\tau$, $I(\tau) = z$, and thus $D(I(\tau), z)$. $\qquad\square$

## B Environments

**Multi-Stage Game** Apart from single-stage game as motivating example, we also conduct experiments also on a multi-stage game in order to highlight the importance of multi-step minimax instead of a single-step minimum by adversary. This is because the optimal adversary will also consider the optimal protagonist strategy in the subsequent states. Considering a multi-step minimax return is more accurate to describe the value of states and sub-trajectories.

**Gambling** Gambling [32] is originally a simple stochastic bandit problem with three arms and horizon one. When taking the first arm, the reward will be either $5$ or $-15$ with $50\%$ probability each. Similarly, when taking the second arm, the reward will be $1$ or $-6$ with equal probabilities. The third arm has deterministic reward $1$. In this paper, we extend it to adversarial setting by considering an adversary controlling the outcomes, shown in Figure 9.

**Connect Four** Connect Four has a long horizon 22 in the worst case. An example of the dynamics of this game is displayed in Figure 8. The outcomes only consist of win, tie, and lose, leading to the reward for the protagonist 1, 0, and $-1$. We leverage the solver in `https://github.com/PascalPons/connect4` to generate stochastic policy for data collection.

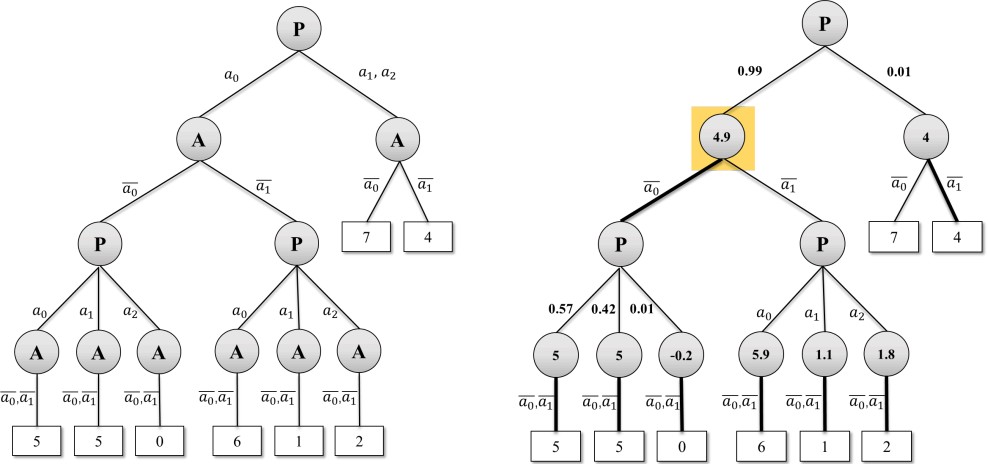

Figure 7: *Multi-stage Game.* **LHS** is the tree representation of the problem, where **P** and **A** are decision maker (protagonist) and adversary, respectively. **RHS** is the same game tree where we attach the learned minimax return and the action probabilities of the well-trained ARDT of a single run when conditioning on the target return 7 to the adversarial node and the branches, respectively. For example, the highlighted node has minimax return-to-go $4.9$. $0.99$ on the branch lead to the highlited node indicates the action probability of taking $a_0$ at the initial state. Thick lines represent the optimal adversarial actions.

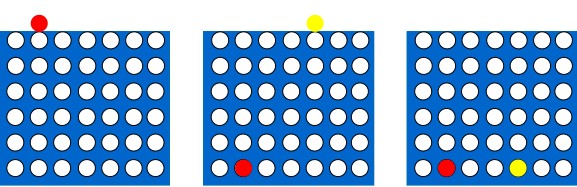

Figure 8: Connect Four game. Two players put pieces to specific columns in turns. The pieces will drop to the final empty row.

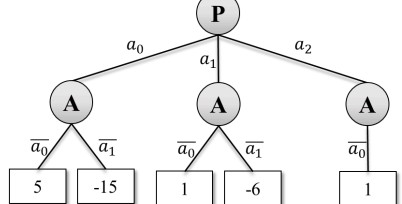

Figure 9: Two-player zero-sum game Gambling: **P**: protagonist, and **A**: adversary. The payoff at the leaves are for protagonist.

## C    Implementation Details

Our implementation is based on the implementation of Decision Transformer [2], and ESPER [32]. Our data collection and part of the evaluation against online adversaries are based on the implementation in Noisy Action Robust Reinforcement Learning [42].

The hyperparameters of the Minimax Expectile Regression are in Table 3. In practical version of ARDT's minimax regression, we adopt a leaf regularization to ensure the accuracy of predicting outcomes at terminal state (leaf node). We record our training iterations of minimax regression in the hyperparameters table. The number of training iterations in full coverage examples and Connect Four are larger than twice of the maximum trajectory length, which guarantees the minimax return propagated from the leaves to every node. According to the prediction in Figure 7, the predicted values are relatively accurate. While in MuJoCo environment, the trajectory length are too long for minimax regression to make full propagation, partial propagate already brings robustness based on the MuJoCo results in Section 4.3.

Decision Transformer is trained with cross entroy loss in discrete environment and MSE in continuous environment, as in Decision Transformer [2]. During action inference at test time, all DT-based methods are prompted with a desired initial target return and state to generate the strategy

| Process | Hyperparameters | Values (Full coverage game/Connect Four/MuJoCo) |
|---|---|---|
| Transformer training | Number of training steps | 5000/100000/100000 |
| | Number of testing iterations | 100 |
| | Context length | 4/20/22 |
| | Learning rate | 0.0001 |
| | Weight decay | 0.0001 |
| | Warm up steps | 1000 |
| | Drop out | 0.1 |
| | Batch size | 128/128/512 |
| | Optimizer | AdamW |
| Minimax expectile regression | Number of training iterations | 6/44/50 |
| | Learning rate | 0.001 |
| | Weight decay | 0.0001 |
| | Model | LSTM or MLP |
| | Batch size | 128/128/512 |
| | Leaf weight | 0.9 |
| | Expectile level | 0.01 |
| | Optimizer | AdamW |

Table 3: Hyperparameters of ARDT.

| Datasets | Returns of Behavior Policy |
|---|---|
| Hopper-lr | 942.8 (417.4) |
| Hopper-mr | 870.5 (467.4) |
| Hopper-hr | 505.4 (532.2) |
| Walker2D-lr | 1330.2 (654.2) |
| Walker2D-mr | 1277.4 (653.9) |
| Walker2D-hr | 1013.1 (581.1) |
| Halfcheetah-lr | 1516.4 (215.8) |
| Halfcheetah-mr | 1395.1 (461.2) |
| Halfcheetah-hr | 795.3 (764.4) |

Table 4: Data profile of MuJoCo NR-MDP.

| Dataset | Returns of Behavior Policy |
|---|---|
| (30, 10) | 0.49 (0.87) |
| (30, 30) | 0.13 (0.99) |
| (30, 50) | -0.23 (0.97) |
| (40, 10) | 0.56 (0.83) |
| (40, 30) | 0.24 (0.97) |
| (40, 50) | -0.13 (0.99) |
| (50, 10) | 0.61 (0.79) |
| (50, 30) | 0.32 (0.95) |
| (50, 50) | -0.02 (1) |

Table 5: Data profile of Connect Four. The tuples represent the optimalities percentages of protagonist and adversary.

$\pi$. Subsequently, the trajectory is generated in an auto-regressive manner, utilizing both the actions sampled by protagonist's strategy $\pi$ and the adversarial strategy $\bar{\pi}$, along with the transitions and rewards provided by the online environments, till the environment terminates or the maximum trajectory length is reached for evaluation. Notably, the target return will be subtracted by the observed immediate reward once received any from the environment.

Notably, given that the rewards and states tokens encompass sufficient information about the adversary's action, we train the Transformer model with sequences without adversarial tokens, as it ensures that the model capacity remains consistent across algorithms. As usual, vanilla DT's is conditioned on the observed returns-to-go, which are contaminated by the adversarial actions, and states. Similarly, ESPER policy is conditional on the expected returns-to-go and states. Since the adversary only influences the transition and rewards, ESPER treats the adversary implicitly part of the environment who causes stochasticity, and addresses it with expected return as the information function.

# D  Additional Related Works

## D.1  Algorithmically Related Works

Tabular value-based works have been studied to solving zero-sum games offline by leveraging pessimism for out-of-sample estimation [4, 56]. In contrast, we did not incorporate pessimism due to the difficulty of tuning its associated hyperparameter. We employ expectile regression to approximate the in-sample minimax return, inspired by the way of Implicit Q Learning [16] to approximate the Bellman optimal $Q$-values leveraging expectile regression (ER). ER is efficient since it doesn't rely

on querying the actions during minimax return estimation, thus reducing complexity. Learning value functions in this way has much less complexity to learn strategy from the data directly. Thus, the protocol of learning values and then extracting policy from them is reasonable. In addition, training Transformer based on relabeled returns-to-go is similar to do policy extraction from the learned values (returns-to-go) [28]. Transformer which demonstrates the strong reasoning in sequence modeling, should be exploited for improving Reinforcement Learning.

## D.2 Related Adversarial Attacks

Within the broad setting of Robust Reinforcement Learning [27], our experiments focus on action-robustness. We conducted experiments on Noisy Action Robust-MDP as a representative environment, following the previous works [7, 42, 14, 34]. According to the different types of attacks proposed in [25, 35], our adversarial perturbation can be viewed as Man In The Middle (MITM) attack formulation proposed in [35], specifically actions attack, as noted in the last paragraph of the Related Work section of [35]. The actions attack we considered in our work can influence all elements in RL from the learner's perspective, including the next state observation, immediate reward, and dynamics, thus potentially causing all types of perturbation introduced in [25, 35]. Therefore, it is reasonable to select actions attack as a representative problem, although our architecture can be extended well to other formulations of robustness to the attacks.

## D.3 Related Robust RL Methods

Robust Reinforcement Learning can be roughly categorized into addressing training-time robustness to poisoning attacks, and test-time robustness to evasion attacks [49]. Poisoning attacks are defined as the manipulation of elements in the training trajectory, including state, actions, and rewards in the data or online environment [25, 35]. The adversary in this paper is similar to the Man In The Middle (MITM) attack formulation in [35]. Robust IQL [55] and RL agent with survival instinct [22] have demonstrated robustness to data poisoning in an offline setting. However, they only aim at the training-time robustness, while testing the algorithms without any attack. Our method is closer to ROLAH [7] and other related works [42, 14, 34], considering adversary appearing in both training-time and test-time serving evasion attacks. Despite the similarity, these methods are all online. Our method learns to achieve the robustness offline, which is more challenging due to potential distribution shift, and the lack of good behavior policy. In this paper, We didn't select offline robust RL as baseline since we only focus on the Reinforcement Learning via Supervised Learning (RvS) methods. We study the robustness of Decision Transformer (DT), and thus provide more analysis on DT-based algorithms including vanila DT and ESPER in various offline learning settings and environments.

# E    Data and Computing Resources

We publish our datasets along with the codebase via `https://github.com/xiaohangt/ardt`. There is no data access restrictions. The Mujoco data profiles are in Table 5 and 4. The MuJoCo data has 1000 number of trajectories, each with 1000 steps of interactions. The Connect Four datasets have also $10^6$ number of steps of interaction in total, where each trajectory has a length at most 22.

We conduct experiments on GPUs: a GeForce RTX 2080 Ti with memory 11GB, and a NVIDIA A100 with memory 80GB. The memory for the entire computing is around 100GB when parallelizing the running over random seeds in Connect Four game, and MuJoCo.

# F    Additional Experimental Results

ARDT can empirically address stochastic environments. Intuitively, this is because by tuning the expectile level , we can balance between expected return and minimax return, as discussed above, to achieve robustness against the distributional environmental changes.

| Datasets | With Adversarial Token | | | Without Adversarial Token | | |
|---|---|---|---|---|---|---|
| | **DT** | **ESPER** | **ARDT** | **DT** | **ESPER** | **ARDT** |
| Hopper-lr | 175.9 (7.5) | 210.6 (21.5) | **302.0 (221.0)** | 112.1 (63.4) | 66.8 (5.5) | **477.9 (219.9)** |
| Hopper-mr | 361.3 (278.0) | 302.8 (159.4) | **454.7 (402.5)** | 330.2 (286.8) | 103.2 (11.7) | **482.2 (83.7)** |
| Hopper-hr | 322.8 (225.9) | 48.8 (0.3) | **158.5 (2.8)** | 298.62 (319.72) | 63.0 (11.5) | **331.69 (187.18)** |
| Walker2D-lr | **475.4 (84.8)** | 406.1 (108.2) | **528.6 (85.7)** | **429.6 (31.9)** | 380.6 (156.9) | 405.6 (39.8) |
| Walker2D-mr | 505.1 (39.5) | 455.2 (47.0) | **485.5 (67.2)** | 391.0 (20.6) | 426.9 (29.7) | **508.4 (61.8)** |
| Walker2D-hr | 511.9 (94.6) | **618.8 (100.5)** | 475.2 (84.2) | 413.22 (54.69) | 502.6 (31.3) | 492.63 (109.28) |
| Halfcheetah-lr | 1170.6 (94.9) | 1368.6 (58.1) | 1323.2 (76.8) | 1416.0 (48.5) | **1691.8 (63.0)** | 1509.2 (33.8) |
| Halfcheetah-mr | **1599.9 (464.2)** | 1433.5 (64.7) | 1299.9 (115.0) | 1229.3 (171.1) | **1725.6 (111.8)** | 1594.6 (137.5) |
| Halfcheetah-hr | 1140.8 (251.4) | **1558.5 (345.5)** | 974.3 (52.3) | 1493.03 (32.87) | **1699.8 (32.8)** | 1765.71 (150.63) |

Table 6: Ablation study on training policy conditioning on adversary tokens with average Worst-case returns against 8 adversaries including algorithms with and without past adversary tokens. ARDT trained without adversarial token has better performance in general.

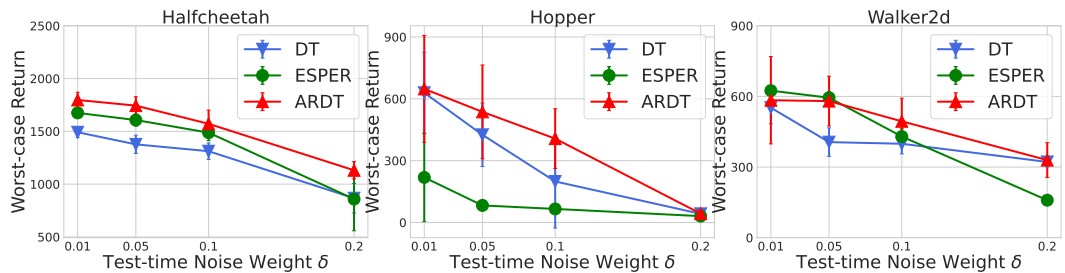

Figure 10: Average results in environments with different test-time weight of adversarial noise. The target return of the three environments are set to 2000, 500 and 800.

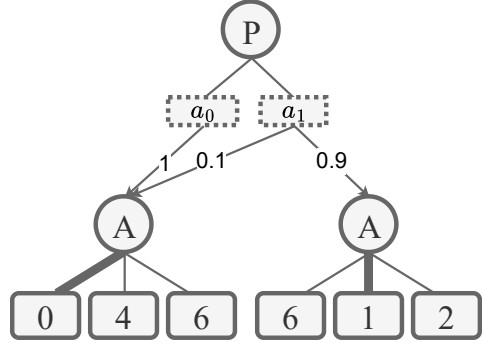

(a) Stochastic Environment revised from Single-stage game, where the protagonist taking action $a_1$ might have probability 0.1 to reach the left branch and 0.9 to reach the right branch.

| Algorithm | Average Return |
|---|---|
| DT | 0.49 (0.0) |
| ESPER | 0.11 (0.2) |
| ARDT (0.3) | 0.23 (0.2) |
| ARDT (0.2) | 0.82 (0.08) |
| ARDT (0.1) | 0.89 (0.01) |
| ARDT (0.05) | **0.91 (0.04)** |
| ARDT (0.01) | 0.9 (0.04) |

(b) Results of DT-based methods on Stochastic Game conditioned on large target return. We vary the expectile level alpha of ARDT and observe that ARDT with a well-tuned alpha can perform the best.

Figure 11: Results on **stochastic environment**.

# G   Additional Broader Impact

When deploying reinforcement learning (RL) methodologies in real-world, safety-critical applications, it is crucial to to account for the worst-case scenarios to prevent unexpected performance drops or safety risks. This approach ensures the system's robustness and reliability under a range of adverse conditions, effectively mitigating the potential for catastrophic failures.

One example is as discussed in the introduction of our paper, in the context of autonomous vehicles, considering worst-case scenarios is essential for safe navigation. The algorithm must be capable of handling sudden changes in road conditions, such as unexpected obstacles or extreme weather, as well as making safe decisions in complex interactive scenarios involving multiple (autonomous) vehicles.

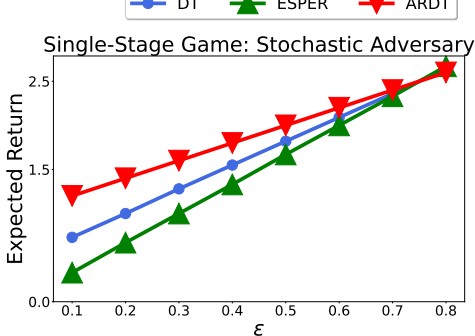

| Algorithm | Average Return |
|---|---|
| DT | 5.19 (0.03) |
| ESPER | 4.67 (0.07) |
| ARDT (0.3) | 4.51 (0.1) |
| ARDT (0.2) | 4.75 (0.32) |
| ARDT (0.1) | 5.2 (0.02) |
| ARDT (0.05) | 5.19 (0.02) |
| ARDT (0.01) | **5.21 (0.04)** |

(a) Expected return computed based on the policy of single random seed in Table 1 in the paper.

(b) Average return conducted on Multi-stage Game, where the adversary has $\epsilon$-greedy policy ($\epsilon = 0.2$). ARDT with a tuned expectile level achieved the best average return.

Figure 12: Results against **stochastic adversary** ($\epsilon$-greedy policy).

| Environment | Algorithm | Normal Return | Worst-case Return | Return Drop |
|---|---|---|---|---|
| Hopper-hr | DT | **809.44 (173.21)** | 298.62 (319.72) | 510.82 |
| Hopper-hr | ESPER | 658.78 (331.68) | 63.0 (11.5) | 595.78 |
| Hopper-hr | ARDT | 746.87 (345.3) | **331.69 (187.18)** | **415.18** |

Table 7: Methods tested in **normal setting**, i.e. without adversarial perturbation.

There are many other real-world control system utilizing RL requires safe and effective operations. In medical applications, such as surgical and assistive robots for patients treatment, preparing for worst-case scenarios—like unexpected drug interactions or patient non-compliance—enhances safety and efficacy. Similarly, in aerospace application, perturbations, such as turbulence, ensures safer aircraft control. Additionally, in the management of plasma states in nuclear fusion reactors using RL methods [6], it is critical to consider the worst-case outcomes of actions to maintain safe and stable operations.

