# OpenReview forum: "Adversarially Robust Decision Transformer"
_NeurIPS.cc/2024/Conference — NeurIPS 2024 poster_

### Official Review · Reviewer_p4QE · 2024-07-13

**Soundness:** 3
**Presentation:** 3
**Contribution:** 3
**Rating:** 6
**Confidence:** 3

**Summary:**

The paper tackles the worst-case-aware RL problem, revises the conventional DT formulation via minimax returns, and proposes the adversarial robust DT to enhance robustness against test-time adversaries.

**Strengths:**

- The formulation of adversarial robust DT with minimax return is sound and clear.
- Improving the robustness against test-time adversaries is critical for real-world deployment of RL.
- The highlight and superiority of the proposed method is verified by comprehensive experiments.

**Weaknesses:**

- Two additional returns-to-go networks are needed for return re-labeling, which increases the computation burden.
- The trade-off between robustness to adversaries and the policy conservativeness is not discussed.

**Questions:**

- Can the ARDT be compared to baselines regarding the normal case, not only the worst case. It could be interesting to see the performance drop in the normal case, and judge where it is worth sacrificing the performance in conventional cases for the improvement in the worst case.
- Are there any real-world applications where it is necessary to consider a worst-case-aware RL algorithm? It could further highlight the significance of the proposed method if some examples could be provided.

**Limitations:**

- The application to real-world scenarios could be limited to some extent, as the learned policy might be too conservative.

---

> ### Author Rebuttal · Authors · 2024-08-07
>
> We thank the reviewer for the insightful comments and suggestions. We address the concern as follows:
>
> ------
>
> **Q: Two additional returns-to-go networks are needed for return re-labeling, which increases the computation burden.**
>
> **A:** The increase in computation is relatively low. And it is worthwhile for the significant benefits it brings, as demonstrated by the performance improvements throughout the experiment section.
>
> - Training returns-to-go networks is a one-time process. Once relabeling is done, the new dataset can be used for training different large models (e.g., Transformer) for all return-conditioned RvS methods. Additionally, since these returns-to-go networks are small models such as MLPs, the GPU memory cost is relatively low.
> - Training small models (e.g., MLPs) to guide the training of large models (e.g., Transformers) is a well-established approach. As we discussed in our related work, both ESPER and Q-Learning DT train small neural networks (e.g. MLP) to predict expected return, and then train DT conditioned on the predicted returns. These methods have shown significant improvements over vanilla Decision Transformer. Apart from Decision Transformer, Diffusion models for control problems also requires additional value networks to guide the training [2,3].
>
> ------
>
> **Q: The trade-off between robustness to adversaries and the policy conservativeness is not discussed.**
>
> **A:** Thanks for this great suggestion! Please refer to our 1st response to reviewer JwFn. We have shown that in Figure 1(c) that  a stochastic adversary and in Figure 2 that a stochastic environment, too much conservativeness does lead to suboptimal performance. In this case, such a trade-off is crucial, while can be addressed by tunning the expectile level $\alpha$.
>
> ------
>
> **Q: Can the ARDT be compared to baselines regarding the normal case, not only the worst case. It could be interesting to see the performance drop in the normal case, and judge where it is worth sacrificing the performance in conventional cases for the improvement in the worst case.**
>
> **A:** Thank you for this suggestion. It is crucial to consider the trade-off between normal performance and achieved robustness with ARDT. We must first clarify the setting in our paper. The assumed performance drop in the normal (non-adversarial) case can only possibly occur when our method is trained with non-perturbed (clean) dataset and tested without attack. While our study assumes that adversarial perturbations are present both during training and testing.
>
> Thus here we only study the performance change of algorithms trained with our adversarial dataset on Hopper, and tested in the normal setting. We provided the experimental results in Table 1 in the rebuttal PDF. To recover the normal case in testing, we set the test-time adversary always taking action zero, i.e. no perturbation on actions. The results show that although ARDT has a lower average return in the normal setting, it demonstrates a smaller performance drop compared to the baselines.
>
> ------
>
> **Q: Are there any real-world applications where it is necessary to consider a worst-case-aware RL algorithm? It could further highlight the significance of the proposed method if some examples could be provided.**
>
> **A:** Thanks for the question! We provide a few examples and will include them in the paper.
>
> When deploying reinforcement learning (RL) methodologies in real-world, safety-critical applications, it is crucial to to account for the worst-case scenarios to prevent unexpected performance drops or safety risks. This approach ensures the system's robustness and reliability under a range of adverse conditions, effectively mitigating the potential for catastrophic failures.
>
> One example is as discussed in the introduction of our paper, in the context of autonomous vehicles, considering worst-case scenarios is essential for safe navigation. The algorithm must be capable of handling sudden changes in road conditions, such as unexpected obstacles or extreme weather, as well as making safe decisions in complex interactive scenarios involving multiple (autonomous) vehicles. There are many other real-world control system utilizing RL requires safe and effective operations. In medical applications, such as surgical and assistive robots for patients treatment, preparing for worst-case scenarios—like unexpected drug interactions or patient non-compliance—enhances safety and efficacy. Similarly, in aerospace application, perturbations, such as turbulence, ensures safer aircraft control. Additionally, in the management of plasma states in nuclear fusion reactors using RL methods [1], it is critical to consider the worst-case outcomes of actions to maintain safe and stable operations.
>
> ------
>
> **Q: The application to real-world scenarios could be limited to some extent, as the learned policy might be too conservative.**
>
> **A:** As we discussed above, we can tune the expectile level parameter alpha to adjust the level of conservativeness. Besides, we also had empirical evidence in our paper indicating that this issue isn't serious. Conservativeness is only problematic when the test-time adversary is weaker than the training-time adversary, causing our method to act suboptimally due to over-conservativeness. While according to Table 1 in our paper, where ARDT is tested under different adversarial perturbation, we can observe that when the adversary is weak as 30% optimal, we can still outperform the baselines.
>
> ------
>
> [1] Degrave, Jonas, et al. "Magnetic control of tokamak plasmas through deep reinforcement learning." *Nature* 602.7897 (2022): 414-419.
>
> [2] Psenka, Michael, et al. "Learning a diffusion model policy from rewards via q-score matching." *arXiv preprint arXiv:2312.11752* (2023).
>
> [3] Wang, Zhendong, Jonathan J. Hunt, and Mingyuan Zhou. "Diffusion policies as an expressive policy class for offline reinforcement learning." *arXiv preprint arXiv:2208.06193* (2022).

---

> > ### Comment · Reviewer_p4QE · 2024-08-10
> >
> > Thank the authors for their detailed response, which has addressed most of my concerns.

---

> > > ### Author Response · Authors · 2024-08-13
> > >
> > > Thank you for your positive response. We are glad that the concerns have been addressed.

---

### Official Review · Reviewer_18ip · 2024-07-14

**Soundness:** 3
**Presentation:** 2
**Contribution:** 3
**Rating:** 5
**Confidence:** 3

**Summary:**

This paper introduces Adversarial Robust Decision Transformer (ARDT), a novel approach enhancing robustness in sequential decision-making. ARDT aligns policies with worst-case scenarios learned through minimax expectile regression, outperforming DT in robustness against powerful adversaries across different data coverage scenarios, including sequential games.

**Strengths:**

1 The paper's motivation seems good and well-founded, and the idea is considered novel and presented clearly and intriguingly.

2 The experimental results demonstrate better robustness compared to existing DT and ESPER methods.

**Weaknesses:**

1 Several attacks on reinforcement learning have been proposed, such as [1][2], why have these not been applied in experimental settings?

2 There have been some related works on robust reinforcement learning, for example [3][4][5]. The authors should provide a comprehensive literature review on the topic of robust reinforcement learning, including similarities, differences, and reasons why it has not been used as a baseline for experimental comparisons.


[1] Optimal Attack and Defense for Reinforcement Learning, AAAI 2024

[2] Understanding the Limits of Poisoning Attacks in Episodic Reinforcement Learning, IJCAI 2022

[3] Towards Robust Offline Reinforcement Learning under Diverse Data Corruption, ICLR 2024

[4] Survival Instinct in Offline Reinforcement Learning, NeurIPS 2024

[5] Robust Reinforcement Learning through Efficient Adversarial Herding, arXiv

**Questions:**

see weaknesses

**Limitations:**

This paper does not consider randomness, whereas random attacks are common in the real world. The authors have already clarified this limitation in the paper.

---

> ### Author Rebuttal · Authors · 2024-08-07
>
> We thank the reviewer for the insightful comments and suggestions. We address the concern as follows:
>
> ------
>
> **Q: Several attacks on reinforcement learning have been proposed, such as [1,2], why have these not been applied in experimental settings?**
>
> **A:** Thanks for the suggested related works. We will include these in the paper along with the next response.
>
> Within the broad setting of Robust Reinforcement Learning [10], our experiments focus on action-robustness.We conducted experiments on Noisy Action Robust-MDP as a representative environment, following the previous works [5-9].  According to the different types of attacks proposed in [1,2], our adversarial perturbation can be viewed as Man In The Middle (MITM) attack formulation proposed in [2], specifically actions attack, as noted in the last paragraph of the Related Work section of [2]. The actions attack we considered in our work can influence all elements in RL from the learner's perspective, including the next state observation, immediate reward, and dynamics, thus potentially causing all types of perturbation introduced in [1,2]. Therefore, it is reasonable to select actions attack as a representative problem, although our architecture can be extended well to other formulations of robustness to the attacks.
>
> ------
>
> **Q: There have been some related works on robust reinforcement learning, for example [3,4,5]. The authors should provide a comprehensive literature review on the topic of robust reinforcement learning, including similarities, differences, and reasons why it has not been used as a baseline for experimental comparisons.**
>
> **A:** Thank you for the great suggestion. We will add the following paragraph in the related work section in the paper to cover the suggested literature:
>
> Robust Reinforcement Learning can be roughly categorized into addressing training-time robustness to poisoning attacks, and test-time robustness to evasion attacks [11]. Poisoning attacks are defined as the manipulation of elements in the training trajectory, including state, actions, and rewards in the data or online environment [1,2]. The adversary in this paper is similar to the Man In The Middle (MITM) attack formulation in [2]. Robust IQL [3] and RL agent with survival instinct [4] have demonstrated robustness to data poisoning in an offline setting. However, they only aim at the training-time robustness, while testing the algorithms without any attack. Our method is closer to ROLAH [5] and other related works [6-9], considering adversary appearing in both training-time and test-time serving evasion attacks. Despite the similarity,these methods are all online. Our method learns to achieve the robustness offline, which is more challenging due to potential distribution shift, and the lack of good behavior policy. In this paper, We didn't select offline robust RL as baseline since we only focus on the Reinforcement Learning via Supervised Learning (RvS) methods. We study the robustness of Decision Transformer (DT), and thus provide more analysis on DT-based algorithms including vanila DT and ESPER in various offline learning settings and environments.
>
> ------
>
> **Q: This paper does not consider randomness, whereas random attacks are common in the real world. The authors have already clarified this limitation in the paper.**
>
> **A:** We do consider random attacks in our paper by modeling an adversary who can have a stochastic policy (see texts below Eq. (1)). We only claim the limitation of deterministic transition, rather than the deterministic policy of adversary.
>
> - In our experiment section 4.2 of our paper, we have experiments against random attack. The test-time adversary attack in our game environment Connect Four is an $\epsilon$-greedy policy (demonstrated in the first paragraph), which is a stochastic policy. The results have shown superior performance than existing DT-based methods.
>
> - To further examine this, we added additional results in the rebuttal PDF Figure 3 against stochastic adversary. According to the figure, ARDT in this single-stage and multi-stage game outperforms the baselines at different levels of stochasticity.
>
> ------
>
> **We appreciate the reviewer's questions and suggestions, and we believe we have addressed all of them, as well as any misunderstandings raised by the reviewer. In light of the reviewer's recognition of the strengths of our work and our detailed rebuttal to their questions, we kindly ask the reviewer to reconsider their score.**
>
> ------
>
> [1] Optimal Attack and Defense for Reinforcement Learning, AAAI 2024
>
> [2] Understanding the Limits of Poisoning Attacks in Episodic Reinforcement Learning, IJCAI 2022
>
> [3] Towards Robust Offline Reinforcement Learning under Diverse Data Corruption, ICLR 2024
>
> [4] Survival Instinct in Offline Reinforcement Learning, NeurIPS 2023
>
> [5] Robust Reinforcement Learning through Efficient Adversarial Herding, arXiv
>
> [6] Tessler, Chen, Yonathan Efroni, and Shie Mannor. "Action robust reinforcement learning and applications in continuous control." International Conference on Machine Learning. PMLR, 2019.
>
> [7] Kamalaruban, Parameswaran, et al. "Robust reinforcement learning via adversarial training with langevin dynamics." Advances in Neural Information Processing Systems 33 (2020): 8127-8138.
>
> [8] Pinto, Lerrel, et al. "Robust adversarial reinforcement learning." *International conference on machine learning*. PMLR, 2017.
>
> [9] Vinitsky, Eugene, et al. "Robust reinforcement learning using adversarial populations." *arXiv preprint arXiv:2008.01825* (2020).
>
> [10] Moos, Janosch & Hansel, Kay & Abdulsamad, Hany & Stark, Svenja & Clever, Debora & Peters, Jan. (2022). Robust Reinforcement Learning: A Review of Foundations and Recent Advances. Machine Learning and Knowledge Extraction. 4. 276-315. 10.3390/make4010013.
>
> [11] Wu, Fan, et al. "Copa: Certifying robust policies for offline reinforcement learning against poisoning attacks." *arXiv preprint arXiv:2203.08398* (2022).

---

> ### Comment · Reviewer_18ip · 2024-08-12
>
> I have read the rebuttal and I will keep my score.

---

> > ### Author Response · Authors · 2024-08-12
> >
> > Thank you for acknowledging our rebuttal! Could you please confirm whether we have addressed all your concerns? Specifically, we would appreciate your feedback on whether the misunderstanding regarding random attacks has been clarified, and the related additional experimental results (Figure 3) in the rebuttal PDF is clear and sensible to you.

---

> > > ### Comment · Reviewer_18ip · 2024-08-13
> > >
> > > The rebuttal clarifies my questions on random attacks. My recommendation is based on my overall evaluation of the work.

---

> > > > ### Author Response · Authors · 2024-08-13
> > > >
> > > > Thank you very much for the reply. We are glad that we managed to clarify and answer all the concerns.

---

### Official Review · Reviewer_JwFn · 2024-07-17

**Soundness:** 3
**Presentation:** 4
**Contribution:** 3
**Rating:** 6
**Confidence:** 3

**Summary:**

This paper consider zero-sum two player Markov game that involves a protagonist and an adversary. The protagonist aims to maximize reward while the adversary aims to minimize it. This paper proposes Adversarial Robust Decision Transformer(ARDT), which is an algorithm that is based on Decision Transformer architecture, and learns the minimax return to go through expectile regression, hence guarantee robustness. Conduct various experiments to validate its algorithm.

**Strengths:**

First to explore the robusteness of reinforcement learning via supervised learning. Great presentation, great novelty. Exhaustive experiement details.

**Weaknesses:**

Lack of ablation study.
e.g. how \alpha can effect the performance?
It seems when \alpha is closer to 1, the learned return to go is closer to the maximum conditioned value,  the more robustness the algorithms should present. It would be more solid to say expetile regression is valid if the experiment can show this phenonmenon.

Like what the author stated in the paper, the environment is all deterministic. It would be great if the experiment can show the algorithm can handle stochastic environment.

line 156 "estiation" to estimation
line 196 "deatils" to "details"

**Questions:**

Updating \omega and \nu happens when another is frozen. Is there any theoretical guarantee this type of update rule can converge to optimal value?

**Limitations:**

The author stated one of their limitations regarding experiments. add experiment that has stochastic transtions if time permit.
Add one more experiment that varies the hyperparameter \alpha to show the effectiveness of expectile regression.

---

> ### Author Rebuttal · Authors · 2024-08-07
>
> We thank the reviewer for the insightful comments and helpful suggestions. We address the concerns as follows:
>
> ------
>
> **Q: Lack of ablation study. e.g. how \alpha can effect the performance? It seems when \alpha is closer to 1, the learned return to go is closer to the maximum conditioned value, the more robustness the algorithms should present. It would be more solid to say expectile regression is valid if the experiment can show this phenomenon.**
>
> **A:** Thank you for pointing this out. We have added an ablation study on $\alpha$ in the rebuttal PDF (Figure 1) and will include these results and analyses in the paper. We varied $\alpha$ for relabeling in Single-stage Game, Multi-stage Game and Connect Four. We tested against the optimal adversary in the first two environments and an $\epsilon$-greedy adversary ($\epsilon=0.5$) for the third environment.
>
> - According to the first two environments, we can confirm a pattern consistent with our theory that smaller alpha leads to more robustness. According to the Equation (6)-(9) in the paper, it should be that when the $\alpha$ is closer to 0, our estimate of minimax return is more accurate, and thus our algorithm is more robust. When the alpha is increased to 0.5, our method is converted to Expected-Return-Conditioned Decision Transformer  (e.g. ESPER) since the expectile loss in this case reduces to Mean Square Error.
> - In the third environment, Connect Four, the performance initially increases as $\alpha$ decreases, but eventually drops. This is due to the weak stochastic adversary. When the $\alpha$ value is too small, the algorithm becomes overly conservative. However, ARDT still outperforms DT significantly. Additionally, it also implies that we can tune the parameter alpha to adjust the level of conservativeness (related to the 2nd and 5th question by reviewer p4QE).
>
> ------
>
> **Q: Like what the author stated in the paper, the environment is all deterministic. It would be great if the experiment can show the algorithm can handle stochastic environment.**
>
> **A:** Thank you for your insightful feedback. We have added a stochastic environment in the rebuttal PDF (Figure 2) and demonstrated that with an appropriate $\alpha$, ARDT can empirically address stochastic environments. Intuitively, this is because by tuning the expectile level $\alpha$, we can balance between expected return and minimax return, as discussed above, to achieve robustness against the distributional environmental changes.
>
> Further Clarification:
>
> - In reinforcement learning (RL) problems with adversarial perturbation, deterministic environments are common. For instance, MuJoCo continuous control, one of the most widely used and standard environments, is employed for experiments in our paper. Even with adversarial perturbation, many MuJoCo tasks still have deterministic transitions. These environments have been extensively used to examine the robustness of algorithms [1-5].
>
> - Moreover, it is natural to adapt our work to directly address stochastic environment by adding an extra value network $V_{\phi}(\tau_{0:t-1}, s)$ in the minimax expectile regression. Instead of minimizing the loss in Eq. (8) and (9) alternatively, we would minimize the loss in Eq. (8) and the following two losses alternatively:
> $$\ell^{1-\alpha}(\omega) = \mathbb{E}\_{\tau \sim \mathcal{D}}[{{Q}\_{\omega}(\tau\_{0:t-1}, {s\_t}, a_t, \bar{a}\_t) - V\_{\widehat{\phi}}(\tau\_{0:t-1}, s\_{t+1}) - r\_t}]^2,\  \ell(\phi) = \mathbb{E}\_{\tau \sim \mathcal{D}} [L^{1-\alpha}\_{\text{ER}}(\widetilde{Q}\_{\widehat{\nu}}(\tau\_{0:t}, {s\_{t+1}},a\_{t+1}) - V\_{\phi}(\tau\_{0:t-1}, s\_{t+1}))].$$ However, this method is just a variant of our minimax expectile regression, so we leave it as a future work.
>
> ------
>
> **Q: Updating \omega and \nu happens when another is frozen. Is there any theoretical guarantee this type of update rule can converge to optimal value?**
>
> **A:** In general, for non-convex-concave optimization problem, the alternative gradient update (fixing \omega and update \mu, vice versa) approach doesn't come with any convergence related theoretical guarantee. However, this is the standard approach used in adversarial ML and robust RL literature, such as GAN for image generation, Robust Adversarial Reinforcement Learning [2,3], and etc [2]. When considering additional structural assumptions on the training objective, some convergence guarantees can be provided [7].
>
> ------
>
> **We believe we have successfully conducted all the additional experiments and addressed all the questions raised by the reviewer. In light of the strengths of our work, as acknowledged by the reviewer, and our detailed rebuttal to the questions posed, we kindly ask the reviewer to reconsider their score.**
>
> ------
>
> [1] Pinto, Lerrel, et al. "Robust adversarial reinforcement learning." *International conference on machine learning*. PMLR, 2017.
>
> [2] Tessler, Chen, Yonathan Efroni, and Shie Mannor. "Action robust reinforcement learning and applications in continuous control." *International Conference on Machine Learning*. PMLR, 2019.
>
> [3] Kamalaruban, Parameswaran, et al. "Robust reinforcement learning via adversarial training with langevin dynamics." Advances in Neural Information Processing Systems 33 (2020): 8127-8138.
>
> [4] Yang, Rui, et al. "Rorl: Robust offline reinforcement learning via conservative smoothing." *Advances in neural information processing systems* 35 (2022): 23851-23866.
>
> [5] Rigter, Marc, Bruno Lacerda, and Nick Hawes. "Rambo-rl: Robust adversarial model-based offline reinforcement learning." *Advances in neural information processing systems* 35 (2022): 16082-16097.
>
> [6] Panaganti, Kishan, et al. "Robust reinforcement learning using offline data." *Advances in neural information processing systems* 35 (2022): 32211-32224.
>
> [7] Mescheder, Lars, Andreas Geiger, and Sebastian Nowozin. "Which training methods for GANs do actually converge?." *International conference on machine learning*. PMLR, 2018.

---

> > ### Comment · Reviewer_JwFn · 2024-08-13
> >
> > Thanks for your explanation and further experiments. It address my concerns and I have add one more point to the rating.

---

> > > ### Author Response · Authors · 2024-08-13
> > >
> > > Thank you for your response and increasing the score. We are pleased that we have been able to clarify and answer all your concerns.

---

### Author Rebuttal · Authors · 2024-08-06

We thank all reviewers for their thoughtful comments and insights. We have revised our manuscript based on your comments and suggestions, and we have responded to each of your individual comments.

## Manuscript Revision Summary

- Typo fixed suggested by reviewer JwFn.
- Added related works suggested by reviewer 18ip.
- Added real-world examples that require considering the worst-case performance, suggested by reviewer p4QE.
- Included all further experiments.

## Further Experiments

- Ablation studies of expectile level $\alpha$ in single-stage game, multi-stage game and Connect Four.
- New results in stochastic environment.
- New results in deterministic games with stochastic adversarial perturbation.
- New results in Hopper tested with no adversarial perturbation .

---

### Decision · Program_Chairs · 2024-09-25

**Decision:**

Accept (poster)

**Comment:**

Interesting piece of work that presents a new algorithm, Adversarial Robust Decision Transformer (ARDT), for scalable offline RL in two-player zero-sum games. Empirical evaluations in full data coverage setting demonstrate the soundness of the proposed method. Good paper.